# Role of WTAP in Cancer: From Mechanisms to the Therapeutic Potential

**DOI:** 10.3390/biom12091224

**Published:** 2022-09-02

**Authors:** Yongfei Fan, Xinwei Li, Huihui Sun, Zhaojia Gao, Zheng Zhu, Kai Yuan

**Affiliations:** 1Department of Thoracic Surgery, The Affiliated Changzhou No. 2 People’s Hospital of Nanjing Medical University, Changzhou 213003, China; 2Heart and Lung Disease Laboratory, The Affiliated Changzhou No. 2 People’s Hospital of Nanjing Medical University, Changzhou 213003, China; 3Department of Gastroenterology, Affiliated Cancer Hospital of Bengbu Medical College, Bengbu 233000, China; 4Department of Radiotherapy, The Affiliated Changzhou No. 1 People’s Hospital of Suzhou University, Changzhou 213003, China

**Keywords:** WTAP, methyltransferase, m^6^A, cancer, molecular mechanisms

## Abstract

Wilms’ tumor 1-associating protein (WTAP) is required for N^6^-methyladenosine (m^6^A) RNA methylation modifications, which regulate biological processes such as RNA splicing, cell proliferation, cell cycle, and embryonic development. m^6^A is the predominant form of mRNA modification in eukaryotes. WTAP exerts m^6^A modification by binding to methyltransferase-like 3 (METTL3) in the nucleus to form the METTL3-methyltransferase-like 14 (METTL14)-WTAP (MMW) complex, a core component of the methyltransferase complex (MTC), and localizing to the nuclear patches. Studies have demonstrated that WTAP plays a critical role in various cancers, both dependent and independent of its role in m^6^A modification of methyltransferases. Here, we describe the recent findings on the structural features of WTAP, the mechanisms by which WTAP regulates the biological functions, and the molecular mechanisms of its functions in various cancers. By summarizing the latest WTAP research, we expect to provide new directions and insights for oncology research and discover new targets for cancer treatment.

## 1. Introduction

Post-transcriptional modifications (PTMs) are an important part of epigenetics that can occur on all types of biological macromolecules, such as RNA, DNA, and proteins, and most PTMs have important effects on physiological processes [1,2]. Currently, more than 170 different RNA chemical modifications have been identified in living organisms [3], including N^6^-methyladenosine (m^6^A), pseudouridine (Ψ), N^4^-acetylcytidine (ac^4^C), 5-methylcytidine (m^5^C), N^1^-methyladenosine (m^1^A), and N^7^-methylguanosine (m^7^G) modifications [2,4]. First discovered in the 1970s [5], m^6^A modification has subsequently been demonstrated to be the most abundant form of internal modification of long noncoding RNAs (lncRNAs) and messenger RNAs (mRNAs) in many eukaryotes, accounting for nearly 80% of RNA methylation modifications [6]. It is estimated that 0.1–0.4% of the total adenine nucleotide content of mammalian transcripts is methylated, particularly at the start of the 3′-UTR, near the translation stop codon, usually embedded in the consensus sequence 5′-RRACH-3′ [7,8,9]. Studies have shown that m^6^A participates in most steps of RNA metabolism, including mRNA transcription, translation, splicing, folding, degradation, and export [9,10]. The completion of these modification processes is dependent on methyltransferases (writers), demethylases (erasers), and methylation-reading proteins (readers). Moreover, the modification process is dynamic and reversible [11,12] (Figure 1). Studies have demonstrated that the m^6^A methylation modifications are involved in the development of various diseases, such as malignant tumors [13,14,15], cardiovascular diseases [16,17], autoimmune diseases [18], and psychiatric diseases [19].

Wilms’ tumor 1-associating protein (WTAP) is a WT-associating protein whose transcription is regulated by WT-1. WTAP is a ubiquitously expressed nuclear protein, and the protein is localized throughout the nucleoplasm as well as in spots, and partially clustered with splicing factor [20]. Studies have demonstrated that WTAP plays an important role in a variety of diseases, such as malignancies [14], cardiovascular diseases [21,22,23], lumbar disk diseases [24], and autoimmune diseases [25]. WTAP regulates the development of these diseases in two forms: one is m^6^A modification dependent on methyltransferase activity and the other is targeted modification independent of m^6^A methyltransferase, of which m^6^A modification dependent on methyltransferase activity is the most widely studied. In addition, WTAP has been demonstrated to catalyze substrate RNAs by forming the methyltransferase-like 3 (METTL3)- methyltransferase-like 14 (METTL14) -WTAP (MMW) complex as a core component of the methyltransferase complex (MTC) [26]. The role of METTL3 and METTL14 as core components of the MTC in tumors has been described in detail [27,28]. In this article, the latest regulatory mechanisms of WTAP in tumors are elaborated to provide researchers with new ideas for tumor treatment.

## 2. Methyltransferase

Methyltransferases, also called ‘writers’, catalyze the m^6^A methylation modification of bases in mRNA [29]. The first methyltransferase was purified as a protein complex by Bokar et al. in 1994 [30]. Further studies revealed an m^6^A MTC with METTL3-METTL14 heterodimer at the core and consisting of other components such as WTAP, vir-like m^6^A methyltransferase associated (VIRMA), zinc finger CCCH-type-containing 13 (ZC3H13), RNA-binding motif protein 15 (RBM15), and RNA-binding motif protein 15B (RBM15B) [31,32,33,34,35,36] (Figure 2). METTL3, the first m^6^A methyltransferase [37], is not only a subunit of the core component of methyltransferases but also a catalytic subunit that uses S-adenosylmethionine (SAM) as a methyl donor [38]. METTL14 was confirmed to be a homologue of METTL3 with 43% homology to METTL3 [39], which forms a stable heterodimer with METTL3 in a 1:1 ratio [31]. METTL3 in the heterodimer plays the catalytic role. Its internal SAM-binding structural domain catalyzes the transfer of methyl groups from SAM to adenine bases in RNA to produce S-adenosyl homocysteine (SAH). In contrast, METTL14 serves mainly to stabilize the structure of MTC and to identify specific RNA sequences 5′-RRACH-3′ as catalytic substrates [32,40,41]. In MTC, WTAP is not catalytically active; however, it promotes m^6^A methylation by binding to METTL3 in the nucleus to form the MMW complex and localizing to the nuclear speckles [32,42]. VIRMA recruits the MMW complex to direct regioselective methylation and mediates preferential mRNA methylation at the 3′UTR, near the stop codon [33]. Very few studies have looked at the role of RBM15 and RBM15B concerning methylation. RBM15 and RBM15B are shown to recruit the MTC to their target transcripts by binding directly to U-rich sequences on mRNA [35]. Further studies showed that ZC3H13 could bind RBM15 to WTAP to catalyze m^6^A methylation [36].

Subsequently, methyltransferase-like 16 (METTL16) [43], methyltransferase-like 5 (METTL5) [44], and CCHC zinc finger-containing protein (ZCCHC4) [45] were found to function as methyltransferases. The N-terminal structural domain of METTL16 with m^6^A methyltransferase activity is responsible for the m^6^A of A43 of the U6 small nuclear RNA (snRNA). The A43 is located in a highly conserved ‘ACAGAGA’ sequence of U6, which is base-paired to 50 splice sites of the pre-mRNA, suggesting an essential role in METTL16 splicing regulation [43,46]. In addition, METTL16 methylation of the ‘UACAGAGA’ sequence in the 3′UTR hairpin of SAM synthase methionine adenosyltransferase 2A (MAT2A), inducing MAT2A splicing to regulate SAM homeostasis [47,48,49]. Recent studies have identified METTL5 as an 18S ribosomal RNA (rRNA) m^6^A-modifying enzyme, and ZCCHC4 was identified as a eukaryotic 28S rRNA m^6^A-modifying enzyme [44,45,50,51].

## 3. WTAP Functions and Its Structural Characteristics

Little et al. first isolated the WTAP protein containing 396 amino acids by a yeast two-hybrid system [20]. Fluorescein in situ hybridization revealed that the WTAP gene is located on human chromosome 6q25–27 and is widely expressed in various human tissues [20]. The structural model of the WTAP protein was generated through the AlphaFold protein structure online website (https://alphafold.ebi.ac.uk/ (accessed on 20 May 2022), Figure 3). Studies have demonstrated that WTAP is involved in m^6^A RNA methylation modification [32,42], cell proliferation [52,53], cell-cycle regulation [34,54,55], RNA splicing [56,57], and embryonic development [34,58,59]. As previously described, WTAP exerts m^6^A modification by binding to METTL3 and METTL14 to form the MMW complex [32]. High-throughput sequencing indicated enriched binding of ‘GGAC’ sequences of METTL3 and METTL14 and ‘GACU’ sequences of WTAP [31]. Interestingly, WTAP did not affect the in vitro activity of the METTL3-METTL14 complex but affected its activity in vivo, independent of WT-1 [42]. Studies have revealed that WTAP consists of an extended N-terminal coiled-coil region followed by a rather unstructured C-terminal [32]. The interaction of WTAP with METTL3 or METTL14 depends on its highly conserved N-terminus [42]. In addition, immunofluorescence microscopy uncovered that METTL3 and METTL14 were present in the nucleus, while WTAP regulated the accumulation of METTL3 and METTL14 in the nuclear speckle [42]. Identification of potential nuclear localization signal (NLS) on METTL3, METTL14 and WTAP revealed how the MMW complex reaches inside the nucleus [32]. WTAP has a potential NLS at its N-terminal end, and METTL3 has a potential NLS in the helical domain between the N-terminal pilot helix and methyltransferase domain (MTD). The NLS of METTL3 and WTAP have nuclear import functionality [32]. NLS in METTL14 is located within its terminal structural domain; however, the NLS in METTL14 does not have the function of transporting METTL14 into the nucleus. Further studies demonstrated that METTL14 localizes to the nucleus by binding to METTL3, forming a heterodimer [32]. However, how WTAP regulates the accumulation of METTL3 and METTL14 in nuclear speckles is yet to be uncovered. mRNA is the major RNA species bound by METTL3 and WTAP. Most binding sites are identified in the coding sequence (CDS), and UTR regions of mRNA and motifs ‘AGGACU’ and ‘UGUGGACU’ are enriched in METTL3 and WTAP-binding clusters, respectively [42]. Additionally, there is a significant spatial correlation with 69% METTL3 and 74% WTAP-binding clusters in UTR and CDS regions overlapping with m^6^A sites [42].

WTAP also acts as a splicing regulator, driving protein diversity and complexity [60]. WTAP is also a component of the interchromatin granule cluster corresponding to nuclear speckles, many of which interact with proteins involved in gene expression, such as pre-mRNA splicing factors and transcription factors [20,61]. A proteomics study identified splicing factors that interact with WTAP, including the proteins thyroid hormone receptor associated protein 3 (THRAP3), BCL2 associated transcription factor 1 (BCLAF1), serine- and arginine-rich splicing factor 1 (SRSF1) and serine- and arginine-rich splicing factor 3 (SRSF3) containing arginine-/serine-rich structural domains and heterogeneous nuclear ribonucleoprotein (hnRNPs), as well as the fragile X mental retardation 1 (FMR1), G-quadruplex-binding proteins, fragile X mental retardation syndrome-related protein 1 (FXR1), and fragile X mental retardation syndrome-related protein 2 (FXR2) [56,57]. In addition, studies have demonstrated that WTAP localizes to nuclear speckles through interactions with THRAP3 and BCLAF1 and colocalizes with splicing factors [20,57,62]. Ping et al. proposed that the MMW complex functions in alternative splicing [42]. However, Horiuchi et al. identified another protein complex including WTAP, VIRMA, Cbl proto-oncogene like 1 (CBLL1), and ZC3H13 in GC-rich sequences, forming G-quadruplexes to inhibit the regulation of alternative splicing and selective polyadenylation [56]. In addition, studies have revealed that WTAP regulates G_2_/M transition through stabilization of cyclin A2 mRNA [55]. WTAP also recruits METTL3 and METTL14 to RNA and actively controls adipogenesis by promoting the mitotic cell-cycle transition. Knockdown of any of the three proteins inhibits the upregulation of cyclin A2 during mitosis, leading to impaired adipogenesis [54]. WTAP knockdown has been shown to promote apoptosis in porcine embryonic cells and affect embryonic development; however, the mechanism remains unclear [59].

## 4. Regulation of WTAP Expression

WTAP is widely expressed in a variety of human tissues (Figure 4) and is dysregulated in cancer expression through different mechanisms (Table 1). Numerous studies have explained the role of different noncoding RNAs (ncRNAs), including microRNAs (miRNAs), lncRNAs, and piwi-interacting RNAs (piRNAs) in tumor development. They have been extensively studied in tumor screening, diagnosis, treatment, prognosis, and drug response [63,64,65]. PiRNA-30473 in diffuse large B-cell lymphoma (DLBCL) reduces WTAP mRNA decline and enhances WTAP mRNA stability by binding to the 3′UTR of WTAP [66]. In renal cancer, miR-501-3p upregulates WTAP expression by targeting binding to WTAP [67]. MiR-139-5p negatively upregulates WTAP expression in hepatocellular carcinoma by targeting the 3′-UTR of WTAP [68]. In addition, the discovery of competing endogenous RNA (ceRNA) regulatory networks further enriched the link between ncRNAs and tumorigenesis [69,70,71,72]. LncRNA PCGEM1 can serve as a ceRNA to compete for miR-433-3 to upregulate WTAP expression in non-small-cell lung cancer (NSCLC) [73]. In hepatocellular carcinoma, LINC00839 upregulates WTAP expression by acting as a ceRNA contending to bind miR-144-3p [74]. Under the hypoxic regime, lncRNA EMS forms a ceRNA to upregulate WTAP expression in esophageal cancer by competing with WTAP to bind miR-758-3p. In addition, overexpression of lncRNA EMS and WTAP in this ceRNA is associated with worse overall survival (OS) and disease-free survival (DFS). In contrast, low expression of miR-758-3p is associated with a poorer DFS and OS. Therefore, ceRNA regulatory network can be used as an indicator for prognosis [75]. LncRNA SNHG10 competes for binding miR-141-3p by acting as a ceRNA to upregulate WTAP expression in osteosarcoma [76]. In pancreatic cancer, lncRNA DUXAP8 competitively binds miR-448 to upregulate WTAP expression [77]. Interestingly, WTAP promotes the stability of lncRNA DLGAP1-AS1 through m^6^A modification, and lncRNA DLGAP1-AS1 upregulates WTAP expression in breast cancer through 3′-UTR binding to miR-299-3p. However, whether lncRNA DLGAP1-AS1 competes with WTAP to bind miR-299-3p remains to be understood [78].

There have been studies identifying ncRNAs independent mechanisms for regulation of WTAP expression. Heat shock protein 90 (Hsp90) can stabilize WTAP protein expression by inhibiting the ubiquitin–proteasome pathway [79]. In pancreatic cancer, overexpression of pseudogene Wilms tumor 1-associated protein pseudogene 1 (WTAPP1) enhances the recruitment of more translation initiation factor–eukaryotic translation initiation factor 3 (EIF3) complex (EIF3A to EIF3M) by EIF3B and promotes WTAP translation [80]. Inhibitor of growth family member 2 (ING2) was shown to negatively regulate WTAP expression in NSCLC [81]. In colorectal cancer, increased expression of arrestin beta 2 (ARRB2) could upregulate WTAP expression; however, the specific regulation mechanism is unclear [82]. In breast cancer, extracellular signal-regulated kinase 1 (ERK1) and extracellular signal-regulated kinase 2 (ERK2) phosphorylate WTAP at serine 341 and stabilize WTAP, thus upregulating WTAP protein levels [83]. In nasopharyngeal carcinoma, WTAP is upregulated by CREB-binding protein (KAT3A)-mediated acetylation of H3K27 [84]. He et al. analyzed glioma single-nucleotide polymorphisms in the Chinese child population and found that WTAP rs7766006 increased the risk of glioma by upregulating WTAP mRNA expression; however, these findings were preliminary and lacked reliable experimental support [85]. In Epstein–Barr virus-associated gastric carcinoma (EBVaGC), EBV-encoded small RNA1 (EBER1) can downregulate WTAP expression by activating the nuclear factor kappa-B (NF-κB) signaling pathway [86]. In addition, the mechanistic target of rapamycin complex 1 (mTORC1) signaling has also been shown to increase WTAP protein abundance [87]. Subsequently, Cho et al. revealed that mTORC1 via its kinase S6K enhanced the targeting of 5′UTR of eIF4A/4B to WTAP mRNA, leading to the translation of WTAP [52]. Interestingly, METTL3, one of the methyltransferase complex members, is crucial for WTAP homeostasis, and both its knockdown and overexpression can lead to the upregulation of WTAP protein. On the one hand, increased METTL3 levels can lead to higher WTAP protein levels, independent of the METTL3 catalytic activity but by increasing WTAP mRNA translation and protein stability. On the other hand, decreased METTL3 levels lead to increased WTAP mRNA levels, ultimately leading to increased WTAP protein levels. However, in the absence of functional METTL3, upregulation of WTAP was not sufficient to promote cell growth [88].

**Table 1 biomolecules-12-01224-t001:** Regulation of WTAP expression.

Type	Moleculars/Signals	Mechanism	Expression	Tumor Types	Reference
piRNA	piRNA-30473	Targeted WTAP	Enhancing the stability of WTAP mRNA	DLBCL	[66]
miRNAs	miR-501-3p	Targeted WTAP	Upregulation of WTAP expression	Renal cancer	[67]
	miR-139-5p	Targeted WTAP	Upregulation of WTAP expression	Hepatocellular carcinoma	[68]
lncRNAs	lncRNA PCGEM1	lncRNA PCGEM1/miR-433-3/WTAP	Upregulation of WTAP expression	NSCLC	[73]
	LINC00839	LINC00839/miR-144-3p/WTAP	Upregulation of WTAP expression	Hepatocellular carcinoma	[74]
	lncRNA EMS	lncRNA EMS/miR-758-3p/WTAP	Upregulation of WTAP expression	Esophageal cancer	[75]
	lncRNA SNHG10	lncRNA SNHG10/miR-141-3p/WTAP	Upregulation of WTAP expression	Osteosarcoma	[76]
	lncRNA DUXAP8	lncRNA DUXAP8/miR-448/WTAP	Upregulation of WTAP expression	Pancreatic cancer	[77]
	lncRNA DLGAP1-AS1	lncRNA DLGAP1-AS1/miR-299-3p/WTAP	Upregulation of WTAP expression	Breast cancer	[78]
Genes	Hsp90	Inhibiting the ubiquitin–proteasome pathway	Stabilization of WTAP expression	DLBCL	[79]
	WTAPP1	Recruiting more translation initiation factor EIF3 complex to promote WTAP translation	Upregulation of WTAP expression	Pancreatic cancer	[80]
	ING2	Negative regulation of WTAP expression	Upregulation of WTAP expression	NSCLC	[81]
	ARRB2	Interaction with WTAP	Upregulation of WTAP expression	Colorectal cancer	[82]
	ERK1/ERK2	Phosphorylation of WTAP at serine 341 and stabilization of WTAP	Upregulation of WTAP expression	Breast cancer	[83]
	KAT3A	KAT3A-mediated acetylation of H3K27	Upregulation of WTAP expression	Nasopharyngeal carcinoma	[84]
	EBER1	Activation of NF-κB signaling pathway	Downregulation of WTAP expression	EBVaGC	[86]
	METTL3	Increased METTL3 levels promote translation of WTAP mRNA and protein stabilization	Upregulation of WTAP expression	/	[88]
		Decreased METTL3 levels lead to increased WTAP mRNA levels	Upregulation of WTAP expression	/	[88]
Other forms	rs7766006	single-nucleotide polymorphisms	Upregulation of WTAP expression	Glioma in Chinese children	[85]
	mTORC1-S6K pathway	Enhanced eIF4A/4B-targeted WTAP translation	Upregulation of WTAP expression	/	[52]

piRNAs: piwi-interacting RNAs; miRNAs: microRNAs; lncRNAs: long noncoding RNAs; DLBCL: diffuse large B-cell lymphoma; NSCLC: non-small-cell lung cancer; EBVaGC: Epstein–Barr virus-associated gastric carcinoma.

## 5. Role of WTAP in Carcinogenesis

With these critical biological and pathological functions, WTAP has been increasingly studied for its crucial role in various tumors, both dependent and independent of its methyltransferase activity (Table 2).

### 5.1. Blood System

#### 5.1.1. Acute Myeloid Leukemia

Acute myeloid leukemia (AML) is one of the most common malignancies of the hematopoietic system [89]. Previous studies indicated that WTAP is overexpressed in AML patients and can be an independent risk prognostic factor for AML [90]. In vitro studies demonstrated that the knockdown of WTAP reduces the proliferative capacity of AML cells, shifts the cell cycle to the G1/S phase, and makes the cells more susceptible to apoptosis [90]. In addition, WTAP overexpression leads to chemotherapeutic agents (Rifamycin) resistance in AML cells [90]. Further studies have demonstrated that WTAP knockdown in AML prolongs the half-life of MYC mRNA by decreasing the level of m^6^A methylation, and thereby upregulates MYC expression to activate the phosphatidylinositol 3-kinase (PI3K)/protein serine-threonine kinase (AKT) signaling pathway [90]. However, the consequences of PI3K/AKT signaling regulation by WTAP on cell behaviors in AML are yet to be characterized [90].

MiR-550-1 expression is downregulated in AML and predicts poor prognosis [91]. Hu et al. used a public dataset and in silico analysis to infer that WTAP is a direct target gene downstream of miR-550-1 [91]. Further analysis suggested that miR-550-1 may mediate a reduction in m^6^A levels by targeting WTAP, which further reduces WW domain containing transcription regulator 1 (WWTR1) stability and mediates, at least in part, its antileukemic activity [91].

#### 5.1.2. Lymphoma

Lymphoma is a hematologic disorder manifested by the clonal proliferation of lymphocytes. DLBCL is the most common pathological subtype of lymphoma, accounting for approximately 25% to 30% of patients with lymphoma [92]. WTAP is overexpressed in DLBCL and can promote cell proliferation and inhibit apoptosis [79]. Treatment with the antitumor drug (Etoposide) in WTAP-silenced cells induced an increased apoptosis rate. The researchers speculated that DLBCL patients with high WTAP expression levels might experience ineffective treatment with chemotherapy [79]. Previous studies have indicated that Hsp90 stabilizes many tumor-promoting proteins [93]. Hsp90 has also been shown to stabilize WTAP expression by inhibiting the ubiquitin–proteasome pathway [79]. In addition, B-cell lymphoma 6 (BCL6), a transcriptional repressor, a vital tumor protein in DLBCL that has been evaluated as a therapeutic target, forms a complex with Hsp90 and WTAP [94]. However, whether the WTAP-Hsp90-BCL6 complex is involved in the transcriptional activity or hinders other cellular functions has not been further explored [79].

WTAP expression was enhanced in human NK/T-cell lymphoma (NKTCL) cells. In vitro experiments showed that WTAP deletion inhibited proliferation and antitumor drug (cisplatin) resistance of NKTCL cells and promoted apoptosis of tumor cells. Mechanistically, WTAP stabilizes dual-specificity phosphatase 6 (DUSP6) mRNA in an m^6^A-dependent manner and targets it for mRNA methylation to confer NKTCL chemoresistance [95].

### 5.2. Digestive System

#### 5.2.1. Esophageal Cancer

Esophageal cancer often has poor prognosis due to its aggressive nature, and is reported to be the sixth leading cause of cancer-related death worldwide [96]. WTAP, an oncogene, is overexpressed in esophageal cancer cells [75,97]. In addition, WTAP upregulation in esophageal cancer is associated with poor prognosis [97]. Higher WTAP expression was observed in esophageal squamous cell carcinoma compared to esophageal adenocarcinoma [97]. Studies have exhibited that hypoxia generates intratumoral oxygen gradients that contribute to tumor plasticity and heterogeneity and promote tumor cell invasion, metastasis, and drug resistance [98]. WTAP overexpression under normoxic and hypoxic conditions promoted cell proliferation, inhibited apoptosis, and induced resistance to cisplatin [75]. Mechanistically, lncRNA EMS is overexpressed in hypoxic environment and can act as ceRNA to compete for binding miR-758-3p to upregulate WTAP expression to promote cisplatin resistance in esophageal tumor cells [75].

#### 5.2.2. Gastric Cancer

Gastric cancer is the fifth most common cancer and the third most common cause of cancer death globally [99]. WTAP was overexpressed in gastric cancer and is associated with poor prognosis [100,101,102], and accelerates glucose uptake and promotes lactate production in gastric cancer cells [100]. WTAP overexpression also accelerates the extracellular acidification rate of gastric cancer cells [100]. These studies suggest that WTAP promotes the Warburg effect in gastric cancer cells [100]. Furthermore, silencing WTAP leads to the inhibition of proliferation in gastric cancer cells [100]. Mechanistically, WTAP promotes the stability of hexokinase 2 (HK2) mRNA, a core molecule that regulates glucose metabolism, thus increasing glucose uptake in gastric cancer cells [100]. In addition, the bioinformatic analysis showed that high expression of WTAP is associated with RNA methylation, while low expression of WTAP is associated with high T-cell-related immune responses [101].

#### 5.2.3. Liver Cancer

Liver cancer is the sixth leading primary cancer and the fourth leading cause of cancer-related deaths worldwide [103]. WTAP is overexpressed in hepatocellular carcinoma and predicts poor prognosis [53]. Cellular and bioinformatics studies showed that the growth and proliferation of cancer cells are inhibited after WTAP knockdown [53,104]. Mechanistically, WTAP inhibits the expression of ETS proto-oncogene 1 (ETS1), a tumor suppressor gene, in an m^6^A HuR-mediated manner. WTAP knockdown led to a significant decrease in m^6^A levels, while the expression levels of ETS1 were increased. ETS1 silencing rescued the inhibitory effect of WTAP knockdown on the growth and viability of hepatocellular carcinoma cells. Interestingly, further studies established that WTAP inhibits ETS1 expression by interfering with the link between HuR protein and ETS1 mRNA [53]. In addition, WTAP knockdown led to a G2/M phase block in hepatocellular carcinoma via the ETS1-p21/p27 axis, promoting apoptosis in tumor cells [53]. WTAP knockdown inhibited hepatocellular carcinoma growth and metastasis in hepatocellular carcinoma by promoting autophagosome production [105]. Further mechanistic studies revealed that liver kinase B1 (LKB1), the upstream kinase of Adenosine 5′-monophosphate activated protein kinas (AMPK), is regulated by WTAP and mediates AMPK phosphorylation in an m^6^A-dependent manner, thereby promoting hepatocellular carcinoma autophagy [105,106,107].

Several studies showed that ncRNAs interact with WTAP to influence hepatocellular carcinoma progression. LINC00839 upregulates WTAP expression and enhances the malignant features of hepatocellular carcinoma by regulating the miR-144-3p/WTAP axis. The proliferation of hepatocellular carcinoma cells was inhibited when LINC00839 was knocked down, and overexpression of WTAP expression eliminated this functional inhibition [74]. WTAP overexpression in hepatocellular carcinoma maintained the malignant phenotype of hepatocellular carcinoma. Further studies demonstrated that miR-139-5p targets the 3′-UTR of WTAP to promote the proliferation and invasive ability of hepatocellular carcinoma by regulating epithelial–mesenchymal transition (EMT) signaling [68].

#### 5.2.4. Pancreatic Cancer

Pancreatic cancer is the most malignant tumor, with a 5-year survival rate of about only 10% [108]. Overexpression of WTAP promoted migration, invasion, and gemcitabine resistance in pancreatic cancer cells [109]. Further studies uncovered that WTAP directly bound focal adhesion kinase (Fak) mRNA and stabilized it, activating the Fak signaling pathway and promoting the malignant characteristics of pancreatic cancer cells. A small-molecule Fak inhibitor (GSK2256098) was able to reverse the cancer-promoting effect of WTAP on pancreatic cancer [109]. LncRNA DUXAP8 can regulate the Fak signaling pathway through miR-448/WTAP axis to promote migration and invasion of pancreatic cancer cells [77]. WTAPP1 was significantly elevated in pancreatic ductal adenocarcinoma and is associated with a poor patient prognosis, and promotes the proliferation and invasive ability of the cells. Mechanistically, increased WTAPP1 mRNA levels in pancreatic ductal adenocarcinoma are caused by m^6^A modification, which stabilizes WTAPP1 mRNA by recruiting the RNA stabilizer HuR by CCHC-type zinc finger nucleic-acid-binding protein (CNBP). Overexpressed WTAPP1 mRNA binds WTAP mRNA and enhances the recruitment of the transcription factor EIF3 complex, and further promotes WTAP translation [80]. Ultimately, the increase in WTAP protein enhanced the activation of the Wnt signaling and stimulated the malignancy of pancreatic ductal cells [80].

#### 5.2.5. Colorectal Cancer

Colorectal cancer (CRC) is the fourth most common malignancy globally. Many studies have shown its close association with epigenetic dysregulation [110]. WTAP has significantly higher expression in CRC than in colorectal adenomas and normal tissue. However, survival analysis revealed that there is no significant difference between high and low WTAP expression in CRC [111]. However, in contrast to Dong et al., Wang et al. demonstrated that high WTAP expression predicted poor prognosis from the perspective of tumor differentiation [112]. The degree of tumor differentiation refers to the degree of cell maturation. Highly differentiated tumor cells are relatively less malignant, while undifferentiated or poorly differentiated tumor cells are relatively more malignant. At the transcriptional and translational levels, WTAP, METTL3, and METTL14 were highly expressed in hypodifferentiated tumor tissues, and the total m^6^A expression level was generally higher in general hypodifferentiated tissues. This suggested that a high total m^6^A level and higher expression of WTAP, METTL3, and METTL14 may be a malignant feature [112]. In addition, bioinformatics analysis inferred that WTAP expression was significantly higher in colon adenocarcinoma than in normal tissue, while no significant difference was found in rectal adenocarcinoma [113].

Carbonic anhydrase IV (CA4) is downregulated in CRC, and upregulating CA4 in in vitro experiments inhibits CRC cell proliferation, induces apoptosis, and stalls the cell cycle in G1 phase [114]. Mechanistically, CA4 interacts directly with WTAP, and the tumor suppressor function of CA4 is dependent on WTAP modification. WTAP knockdown eliminates the inhibitory effect of CA4 on CRC cell viability and colony-forming ability. In addition, CA4 can inhibit CRC progression by inducing WTAP degradation through polyubiquitination, and CA4 also inhibits the Wnt/β-linked protein signaling pathway by activating the WTAP degradation product WT-1 [114]. Urothelial cancer associated 1 (UCA1) was also upregulated in CRC and promoted tumor cell proliferation. It was later discovered that METTL3 and WTAP could m^6^A modify UCA1, and METTL3 and WTAP knockdown reduced UCA1 expression and inhibited CRC cell proliferation [115]. In addition, overexpression of ARRB2 led to the upregulation of WTAP, which promoted CRC cell proliferation and migration. However, the specific regulatory mechanism remains to be understood [82].

#### 5.2.6. Cholangiocarcinoma

Cholangiocarcinoma is a highly lethal epithelial malignancy that is highly aggressive and heterogeneous [116]. Immunohistochemistry showed that WTAP was overexpressed in cholangiocarcinoma tissues and promoted the proliferation, invasion, and migration of cholangiocarcinoma cells. In addition, WTAP overexpression in cholangiocarcinoma induced the expression of metastasis-related genes such as cathepsin H, matrix metallopeptidase 7 (MMP7), matrix metallopeptidase 28 (MMP28), and mucin 1 (Muc1); however, the specific regulatory mechanisms have not been investigated [117].

### 5.3. Reproductive System

#### 5.3.1. Breast Cancer

Breast cancer is the most prevalent malignancy in women and is the leading cause of most cancer-related deaths among women [118]. C5aR1^+^ neutrophil overexpression in breast cancer is associated with poor survival. C5aR1^+^ also induces breast cancer glycolysis. Further studies inferred that tumor necrosis factor-α (TNFα) and interleukin-1β (IL1β) secreted by C5aR1^+^ neutrophils synergistically activated the ERK1/2 signaling to phosphorylate WTAP at serine 341, thereby stabilizing WTAP, which further promotes RNA m^6^A modification of enolase 1 (ENO1) and leads to upregulation of ENO1, thus affecting the glycolytic activity of breast cancer cells [83]. LncRNA DLGAP1-AS1 is highly expressed in Adriamycin-resistant breast cancer cells and promotes the proliferation of drug-resistant cells. Mechanistically, WTAP promotes the stability of lncRNA DLGAP1-AS1 in breast cancer through m^6^A modifications. Interestingly, overexpression of lncRNA DLGAP1-AS1 3′-UTR targeted miR-299-3p to upregulate WTAP expression. However, further exploration is required to understand the lncRNA DLGAP1-AS1/miR-299-3p/WTAP regulatory axis in breast cancer [78].

#### 5.3.2. Endometrial Cancer

Endometrial cancer is the most common gynecological cancer in developed countries, and its incidence is increasing worldwide [119]. WTAP is overexpressed in endometrial cancer. In vitro experiments revealed that WTAP deletion enhanced cisplatin-induced apoptosis and induced cell-cycle arrest. Further analysis revealed that WTAP deletion led to increased expression of the proapoptotic proteins BCL2-associated X (BAX) and cleaved poly ADP-ribose polymerase (PARP) and decreased expression of the antiapoptotic protein myeloid cell leukemin-1 (Mcl-1), and induced cell cycle G2/M phase arrest [120]. Mechanistic analysis indicated that WTAP induced resistance to cisplatin by promoting glycogen synthase kinase 3β (GSK3β) phosphorylation at GSK-3β (Ser9 site) and by activating the Wnt/β-catenin pathway by promoting β-catenin translocation to the nucleus [120]. A high WTAP expression in endometrial cancer is closely associated with its poor prognosis, and it promotes endometrial cancer cell proliferation, migration, and invasion and inhibits apoptosis [121]. Mechanistic studies showed that WTAP increased the m^6^A modification of caveolin 1 (CAV-1) mRNA 3′-UTR and downregulated CAV-1 expression, thereby activating the NF-κB signaling pathway in endometrial cancer to maintain the malignant characteristics of tumor cells [121].

#### 5.3.3. Ovarian Cancer

Ovarian cancer is the third most common gynecological malignancy globally, but has the highest mortality rate among gynecological tumors [122]. WTAP overexpression in ovarian cancer is associated with poor prognosis. In vitro experiments revealed that proliferation, invasion, and migration of tumor cells were inhibited in WTAP knockdown [123]. Furthermore, HBS1-like translational GTPase (HBS1L) and family with sequence similarity 76 member A (FAM76A) were identified as two WTAP-related hub genes by the weighted gene coexpression network analysis method, and WTAP-mediated m^6^A modifications may regulate them. However, this in silico conclusion needs further experimental validation [123]. WTAP was also overexpressed in high-grade serous ovarian cancer and correlated with poor prognosis. WTAP knockdown led to the inhibition of tumor cell proliferation and promotion of apoptosis. Moreover, WTAP downregulation increased E-calmodulin expression and decreased waveform protein expression, thus inhibiting migration [124]. Mechanistically, WTAP maintains the high-grade serous ovarian cancer phenotype by associating with MAPK and AKT signaling pathways; however, further experimental validation is required [124].

### 5.4. Urinary System

#### 5.4.1. Renal Cell Carcinoma

Renal cell carcinoma accounts for more than 400,000 new patients worldwide each year, with the majority of patients being elderly men over 60 years of age [125]. WTAP is upregulated in renal cell carcinoma and is associated with poor prognosis. In vitro experiments implied that high WTAP expression could promote the migration and proliferation of renal cell carcinoma [126]. Further studies revealed that WTAP enhanced cyclin dependent kinase 2 (CDK2) mRNA stability by directly targeting to the 3′-UTR of CDK2 transcript, thus upregulating CDK2 expression and promoting the proliferation of renal cell carcinoma cells [126]. Downregulation of miR-501-3p expression in renal cell carcinoma promotes renal cancer cell proliferation [67]. Mechanistically, miR-501-3p negatively regulates WTAP expression in renal cell carcinoma by targeting the 3′-UTR of WTAP [67]. In flow cytology experiments, it was observed that WTAP knockdown resulted in many G1/S phase blocks, suggesting that WTAP downregulation inhibits the proliferation of kidney cancer cells [67]. Additionally, miR-501-3p overexpression or WTAP knockdown altered the overall RNA m^6^A levels in renal cancer cells. However, the effect on renal cancer cells remains to be explored [67].

#### 5.4.2. Bladder Cancer

Bladder cancer is a common malignancy in women and is the fourth most common malignancy in men [127]. WTAP expression was significantly increased at the transcriptional and translational levels than in normal tissues. Its overexpression is associated with poor prognosis [128,129]. Moreover, knockdown of WTAP in bladder cancer cells significantly accelerated tumor cell apoptosis and attenuated cell viability and the ability of anchored non-dependent growth [129]. Further studies indicated that circ0008399 and WTAP colocalized and interacted in cells, but their expression did not affect each other’s expression [129]. Mechanistically, circ0008399 promotes the formation of MMW complex to promote the overall level of m^6^A modification in bladder cancer cells. circ0008399/WTAP promotes TNFAIP3 expression by increasing m^6^A-dependent TNF alpha induced protein 3 (TNFAIP3) mRNA stability in bladder cancer cells, which in turn inhibits bladder cancer cell apoptosis and mediates reduced bladder cancer chemosensitivity to cisplatin [129]. Additionally, WTAP overexpression could impair the promotion of bladder cancer cell apoptosis induced by circ0008399 knockdown, as well as reverse the reduction in viability and anchorage-independent growth ability [129].

### 5.5. Respiratory System

#### 5.5.1. Nasopharyngeal Cancer

Nasopharyngeal cancer has a distinct geographic distribution and is particularly prevalent in East and Southeast Asia, closely related to lifestyle and environment [130]. In nasopharyngeal carcinoma, KAT3A upregulates WTAP expression by mediating H3K27 acetylation, and WTAP overexpression predicts poor prognosis and promotes tumor cell growth and metastasis [84]. Mechanistic studies have indicated that lncRNA DIAPH1-AS1 was identified as an m^6^A target of WTAP. WTAP enhances its stability by mediating m^6^A modification of lncRNA DIAPH1-AS1 through synergistic interaction with insulin-like growth factor 2 mRNA-binding protein 2 (IGF2BP2) [84]. Furthermore, lncRNA DIAPH1-AS1 acts as a molecular adapter to promote the formation of lncRNA MTDH-LASP1 complex and upregulate LIM and SH3 protein 1 (LASP1) expression, ultimately promoting the growth and metastasis of nasopharyngeal carcinoma cells [84].

#### 5.5.2. Lung Cancer

Lung cancer has the highest mortality rate of all cancers, with an estimated 2 million new cases and 1.76 million deaths per year [131]. ING2 is downregulated in NSCLC tissues. Cell function analysis indicated that overexpressing ING2 inhibited the growth, infiltration, and metastasis, and blocked the apoptosis of NSCLC cells through EMT signaling. WTAP expression is inhibited by overexpressed ING2. Rescue experiments show that WTAP downregulation and overexpression partially mitigate the effects of ING2 knockdown and overexpression on tumor cell proliferation, respectively [81]. However, further analysis of the specific regulatory mechanisms between WTAP and EMT is yet to be unraveled [81]. Weng et al. found that the lncRNA PCGEM1 was overexpressed in NSCLC cells and combined with miR-433-3p to upregulate WTAP expression, promoting the NSCLC malignant phenotype [73]. Furthermore, WTAP overexpression reversed the inhibitory effects of lncRNA PCGEM1 silencing on tumor cell proliferation, migration, and invasion [73]. Using bioinformatics, WTAP can be used as an independent prognostic favorable factor for squamous lung cancer; however, validated clinical follow-up data are lacking to support this view [132].

### 5.6. Other Tumors

#### 5.6.1. Osteosarcoma

Osteosarcoma is one of the most common orthopedic malignancies, with a 5-year survival rate of less than 20% after developing metastases [133]. Elevated WTAP in osteosarcoma is associated with poor prognosis and is an independent prognostic risk factor for patients with osteosarcoma [134]. In vitro and in vivo experiments have demonstrated that WTAP acts as an oncogene in osteosarcoma, promoting cell proliferation, invasion, and migration [134]. Further studies have indicated that WTAP inhibits homeobox-containing 1 (HMBOX1) expression by regulating HMBOX1 m^6^A modification at its 3′ UTR in an m^6^A-dependent manner [134]. In osteosarcoma, low HMBOX1 expression predicts poor overall survival and is involved in WTAP-mediated proliferation and metastasis of osteosarcoma. The maintenance of this malignant feature is mediated in part by WTAP/HMBOX1 through the PI3K/AKT signaling pathway [134]. Ren et al. revealed that lncRNA FOXD2-AS1 is overexpressed in osteosarcoma and is associated with poor prognosis. Functional studies have shown that lncRNA FOXD2-AS1 promotes tumor migration, proliferation, and growth in vivo and in vitro [135]. Mechanistically WTAP enhances the stability of lncRNA FOXD2-AS1 through m^6^A modification and upregulates its expression. Overexpression of lncRNA FOXD2-AS1 promotes the progression of osteosarcoma by targeting forkhead box M1 (FOXM1) [135]. Additionally, lncRNA SNHG10 is overexpressed in osteosarcoma, and silencing lncRNA SNHG10 decreases proliferation, migration, and invasion and accelerates apoptosis of osteosarcoma cells. Mechanistically, lncRNA SNHG10 competes as ceRNA to bind miR-141-3p and upregulates WTAP expression to maintain the malignant features of osteosarcoma. lncRNA SNHG10 silencing leads to the inhibition of cell proliferation, migration, and invasion and the promotion of apoptosis. These phenotypes were rescued upon WTAP overexpression [76].

#### 5.6.2. Glioma

The most common malignant primary brain tumor in adults is glioma [136]. WTAP was shown to be overexpressed in glioblastoma [137,138]. In addition, significantly higher WTAP expression is present in high-grade glioma tissues (grades III-IV) than in low-grade glioma tissues (grades I-II). Prognostic analysis showed significantly lower survival in the WTAP high-expression group than in the WTAP low-expression group in low-grade glioma subtypes and high-grade glioma subtypes [138]. Further studies revealed that WTAP overexpression promoted the proliferation, invasion, and migration of glioblastoma cells. Mechanistically, WTAP overexpression enhanced epidermal growth factor signaling-induced EGFR phosphorylation to promote the migration and invasion of glioblastoma cells, whereas WTAP knockdown or overexpression had no significant effect on total EGFR [137]. In addition, dysregulation of WTAP affected the expression of many hub genes involved in cancer cell migration, such as matrix metallopeptidase 3 (MMP3), thrombospondin 1 (THBS1), C-C motif chemokine ligand 2 (CCL2), C-C motif chemokine ligand 3 (CCL3), lysyl oxidase like 1 (LOXL1), and hyaluronan synthase 1 (HAS1) [137].

**Table 2 biomolecules-12-01224-t002:** Molecular regulatory mechanisms of WTAP in tumors.

Cancer Type		Upstream Regulators	Downstream Targets	Mechanism	m^6^A	Target Pathways	Cellular Phenotypes	Reference
Blood system	AML	/	MYC	Affecting the half-life of MYC mRNA	Yes	PI3K/AKT signaling pathway	Proliferation, apoptosis, and drug resistance	[90]
		miR-550-1	WWTR1	Increased WWTR1 stability	Yes	/	Proliferation and apoptosis	[91]
	DLBCL	Hsp90	/	Stabilized WTAP expression	No	/	Proliferation, apoptosis, and drug resistance	[79]
	NKTCL	/	DUSP6	Increased DUSP6 stability	Yes	/	Proliferation, apoptosis, and drug resistance	[95]
Digestive System	Esophageal cancer	lncRNA EMS/miR-758-3p	/	Upregulated WTAP expression	No	/	Invasion, metastasis, and drug resistance	[75]
	Gastric cancer	/	HK2	Increased HK2 stability	Yes	/	Glucose metabolism	[100]
	Liver cancer	/	ETS1	Decreased ETS1 stability	Yes	/	Proliferation and apoptosis	[53]
		/	LKB1	Affecting LKB1 stability	Yes	AMPK signaling pathway	Autophagy	[105]
		LINC00839/miR-144-3p	/	Upregulated WTAP expression	No	/	Proliferation	[74]
		miR-139-5p	/	Upregulated WTAP expression	No	EMT signaling	Proliferation and invasion	[68]
	Pancreatic cancer	/	Fak	Stabilized Fak expression	No	Fak signaling pathway	Migration, invasion, and drug resistance	[109]
		lncRNA DUXAP8/miR-448	/	Upregulated WTAP expression	No	Fak signaling pathway	Proliferation and invasion	[77]
		WTAPP1	/	Increased WTAP stability	Yes	Wnt signaling pathway	Proliferation and invasion	[80]
	CRC	/	CA4	Affecting CA4 stability	No	/	Proliferation and apoptosis	[114]
		CA4	/	Polyubiquitination inhibits WTAP protein degradation	No	Wnt/β-linked signaling pathway	Tumor progression	[114]
		/	UCA1	Affecting UCA1 stability	Yes	/	Proliferation	[115]
		ARRB2	/	/	/	/	Proliferation and migration	[82]
Reproductive system	Breast cancer	IL1β/TNFα activates ERK1/2 signaling	ENO1	Upregulated ENO1 expression	Yes	/	Glycolysis	[83]
		/	lncRNA DLGAP1-AS1	Increased lncRNA DLGAP1-AS1 stability	Yes	/	Proliferation and drug resistance	[78]
		lncRNA DLGAP1-AS1/miR-299-3p	/	Upregulated WTAP expression	No	/	/	[78]
	Endometrial cancer	/	GSK3β	Promoting GSK3β phosphorylation	No	Wnt/β-catenin	Apoptosis and drug resistance	[120]
		/	CAV-1	Decreased CAV-1 stability	Yes	NF-κB signaling pathway	Proliferation, migration, and invasion	[121]
Urinary system	Renal cell carcinoma	/	CDK2	Increased CDK2 stability	No	/	Proliferation and invasion	[126]
		miR-501-3p	/	Upregulated WTAP expression	No	/	Proliferation	[67]
	Bladder cancer	circ0008399	TNFAIP3	Increased TNFAIP3 stability	Yes	/	Apoptosis and drug resistance	[129]
Respiratory system	Nasopharyngeal cancer	KAT3A mediates the acetylation of H3K27	lncRNA DIAPH1-AS1	Increased lncRNA DIAPH1-AS1 stability	Yes	/	Growth and metastasis	[84]
	Lung cancer	ING2	/	Affecting ING2 expression	/	EMT signaling	Proliferation and apoptosis	[81]
		lncRNA PCGEM1/miR-433-3p	/	Upregulated WTAP expression	No	/	Proliferation, migration, and invasion	[73]
Other tumors	Osteosarcoma	/	HMBOX1	Inhibition of HMBOX1 expression	Yes	PI3K/AKT signaling pathway	Proliferation, migration, and invasion	[134]
		/	LncRNA FOXD2-AS1	Increased lncRNA FOXD2-AS1 stability	Yes	/	Proliferation and invasion	[135]
		lncRNA SNHG10/miR-141-3p	/	Upregulated WTAP expression	No	/	Proliferation, migration, and invasion and apoptosis	[76]
	Glioma	/	/	Enhanced EGFR phosphorylation	No	/	Proliferation, migration, and invasion	[137]

AML: Acute myeloid leukemia; DLBCL: diffuse large B-cell lymphoma; NKTCL: NK/T-cell lymphoma; CRC: colorectal cancer; PI3K: phosphatidylinositol 3-kinase; AKT: protein serine-threonine kinase; AMPK: Adenosine 5′-monophosphate activated protein kinas; EMT: epithelial–mesenchymal transition.

## 6. Targeting WTAP for Potential Clinical Prospects

m^6^A modifications are commonly found in eukaryotic transcriptional regulation and are involved in the progression of many cancers, and the use of m^6^A regulators as tumor therapeutic targets provides new ideas for tumor treatment [6,139,140]. mTOR signaling regulates various biological processes such as cell metabolism, immunity, growth, and autophagy [141], and its dysfunction plays an important role in a variety of cancers; thus, mTOR is considered as one of the potential targets for tumor therapy [142,143]. Research has demonstrated that mTOR can regulate overall m^6^A modification levels, and WTAP is required for mTOR-dependent m^6^A modifications [52,87]. Mechanistically, as presented in Table 1, mTORC1 enhances WTAP translation through kinase S6K and regulates cMyc repressor, MAX dimerization protein 2 (MXD2), to inhibit cMyc signaling [52]. As we all know, cMyc is one of the central hubs of tumor progression [144,145]. MXD protein inhibits cMyc activity by competing with the cMyc transcriptional coactivator MAX [146]. Specifically, WTAP inhibits MXD2 expression by m^6^A modification, leading to increased binding of cMyc to MAX, which promotes cMyc transcriptional activity and mTORC1-activated cancer cell proliferation [52]. Recent studies have revealed that METTL3 inhibitors can effectively inhibit the progression of AML, representing the beginning of a new era in cancer treatment [147]. Thus, WTAP inhibitors can increase MXD2 expression, leading to reduced cMyc binding to MAX, thereby inhibiting mTORC1-activated cancer cell proliferation. In summary, the WTAP-MXD2-cMyc axis is a potential therapeutic target for mTORC1-driven cancers (Figure 5).

Chemotherapy and targeted and immune therapies have achieved remarkable results in the clinical treatment of tumors. However, the development of drug resistance has greatly limited the clinical benefits of these treatments [148,149,150]. In recent years, the involvement of m^6^A in tumor regulation has received wide attention, and its induction of tumor drug resistance also provides new directions for tumor therapy [151,152,153]. WTAP enhances the stability of DUSP6 mRNA in an m^6^A-dependent manner, thereby promoting the development of resistance to cisplatin in NKTCL cells [95]. Circ0008399 interacts with WTAP in bladder cancer to increase the stability of TNFAIP3 mRNA in an m^6^A-dependent manner, thereby inducing tumor cell resistance to cisplatin [129]. A growing number of studies suggest that ncRNAs play a key role in tumor drug resistance [154,155,156]. WTAP increases the stability of lncRNA DLGAP1-AS1 in an m^6^A-dependent manner and induces resistance to Adriamycin in breast cancer cells. Meanwhile, upregulated lncRNA DLGAP1-AS1 enhances tumor cell drug resistance through miR-299-3p/WTAP axis feedback [78]. Under hypoxic conditions, the lncRNA-EMS/miR-758-3p/WTAP regulatory axis can induce cisplatin resistance in esophageal cancer cells [75]. In addition, WTAP can directly regulate the formation of tumor drug resistance. WTAP promotes cisplatin resistance in endometrial cancer by promoting GSK3β phosphorylation at Ser9 to activate the Wnt/β-linked protein pathway [120]. In pancreatic cancer, WTAP promotes chemoresistance to gemcitabine by stabilizing Fak mRNA through direct binding to Fak mRNA [109]. In conclusion, WTAP can mediate tumor resistance formation, both dependent and independent of its role in m^6^A modification of methyltransferases. Therefore, WTAP has high clinical therapeutic promise in tumor therapy.

## 7. Conclusions and Perspectives

Malignant tumors have eight biological characteristics, which are resistance to cell death, maintenance of proliferative signals, achievement of replicative immortality, evasion of growth inhibitors, activation of invasion and metastasis, induction/entry into the vasculature, acquiring the ability to avoid immune destruction, and reprogramming of cellular metabolism [157]. Gene epigenetics regulates gene expression mainly at DNA modification, chromatin reorganization, noncoding RNA regulation, and histone modification, which play an essential role in maintaining oncological hallmarks [2,64,158]. With research on enzymes that regulate protein and DNA modifications, it has been discovered that many of these modifying enzymes can be used as targets for cancer therapy [2,139]. m^6^A is the most abundant form of modification on coding and noncoding RNA in eukaryotes [6]. Methyltransferases, such as METTL3 and METTL14, have received much attention because they can catalyze substrate RNA modification, thus participating in tumor development and providing potential targets for tumor therapy [27,28]. WTAP, as one of the core components of MTC, is involved in the development of multiple cancers and is therefore expected to be a potential therapeutic target for tumors.

In this article, we describe the structural characteristics of WTAP and its involvement in regulating the development of various tumors. On the one hand, WTAP acts as a methyltransferase adaptor to catalyze the substrate RNA and affects the stability of RNA, thus participating in the regulation of tumor development; on the other hand, WTAP is involved in tumor development through upstream regulation or targeting of downstream targets. Therefore, WTAP can be considered a core target for tumor treatment. However, there are no reports yet on the regulatory mechanisms of WTAP in tumor immunity, one of the tumor hallmarks. Studies have revealed that the tumor microenvironment of patients can be classified into ‘immunoinflammatory’, ‘immune evasion’, and ‘immunodesert’ based on m^6^A-regulated gene expression by unsupervised clustering analysis [159,160,161]. Different immune-system phenotypes have different immunotherapeutic effects, and choosing the appropriate treatment plan according to different phenotypes is in line with the modern concept of precision therapy. Therefore, it is a good direction for follow-up studies to focus on the role of WTAP in tumor microenvironment.

Additionally, the tumor microenvironment is a complex system, and epigenomics is one of the factors influencing its complexity [157,162]. Studies have established that genes can act as both cancer suppressors and cancer promoters. P53 is an oncogene with a broad and powerful function, and more than half of tumor patients carry P53 mutations, making it a popular target for oncology treatment. However, recent studies have revealed that p53 also possesses a cancer-promoting role [163,164]. Previous articles also advocated that the methyltransferases METTL3 and METTL14 possess both cancer-promoting and cancer-inhibiting effects [27,28]. WTAP, as the main component of methyltransferase complex, has a strong correlation with METTL3 and METTL14; however, we have not found any articles on the role of WTAP as a cancer suppressor, which may be a new direction for future research.

In conclusion, WTAP plays an essential role in various tumors, either dependent or independent of its methyltransferase activity. Research on WTAP is expected to bring new hope for tumor treatment. However, WTAP has been less studied in terms of antitumor immunity, tumor microenvironment, and oncogenic effects, and future studies targeting these aspects would be a good direction.

## Figures and Tables

**Figure 1 biomolecules-12-01224-f001:**
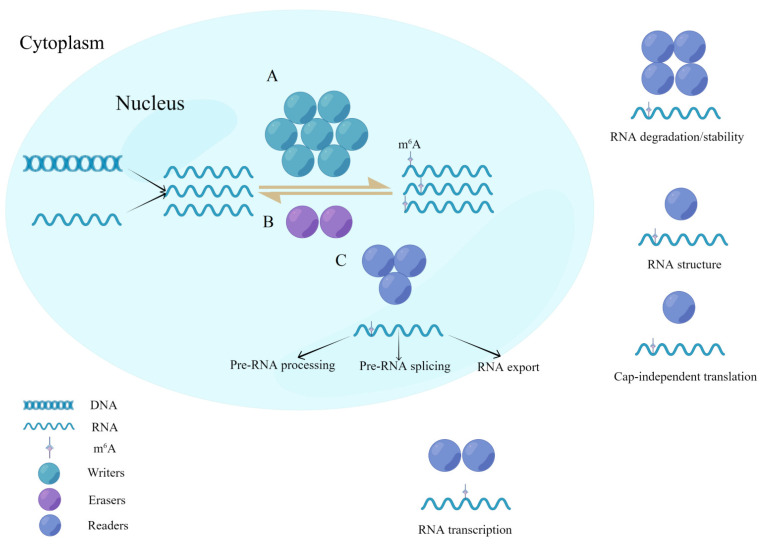
Dynamic reversible process for writers, readers, and erasers in m^6^A modifications. (**A**) Methyltransferases (writers) are a class of catalytic enzymes that catalyze m^6^A methylation modification of bases in substrate RNA by forming a complex. (**B**) Demethylases (erasers) have the opposite function of writers and are capable of demethylating RNA in the nucleus. (**C**) Methylated reading proteins (readers) are specific RNA-binding proteins that undergo m^6^A modifications to produce specific biological functions.

**Figure 2 biomolecules-12-01224-f002:**
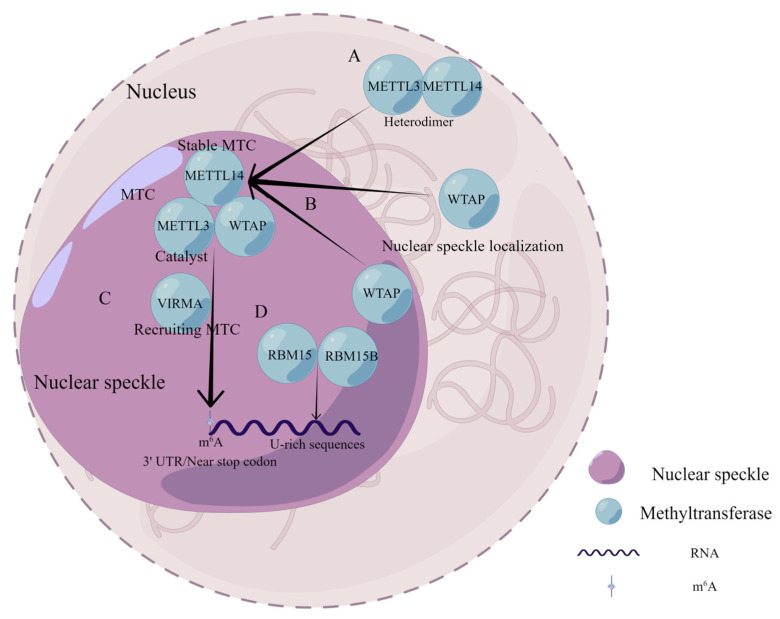
Composition and functional characteristics of MTC in m^6^A modification. (**A**) METTL3 and METTL14 form a stable heterodimer in a 1 to 1 ratio. (**B**) WTAP binds to METTL3 in the nucleus to form the MMW complex and assists in the localization of the complex to the nuclear speckles. (**C**) VIRMA recruits the MMW complex to direct regioselective methylation and mediates preferential mRNA methylation at the 3′UTR, near stop codon. (**D**) RBM15 and RBM15B recruit MTC to their target transcripts by binding directly to U-rich sequences on RNA.

**Figure 3 biomolecules-12-01224-f003:**
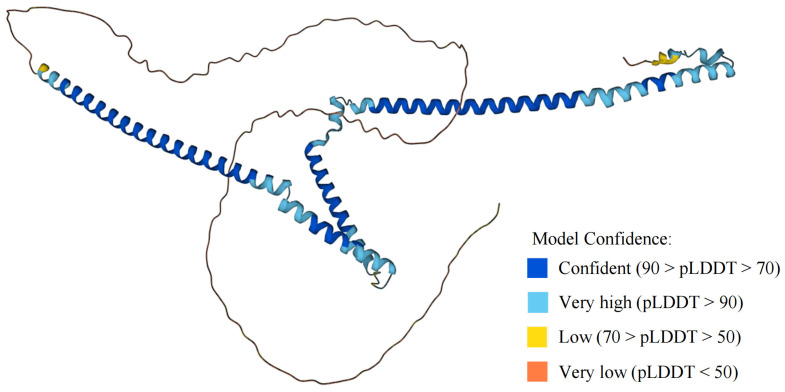
Structural model of WTAP protein in AlphaFold protein structure website. The AlphaFold protein structure website generates a per-residue confidence score (pLDDT) ranging from 0 to 100.

**Figure 4 biomolecules-12-01224-f004:**
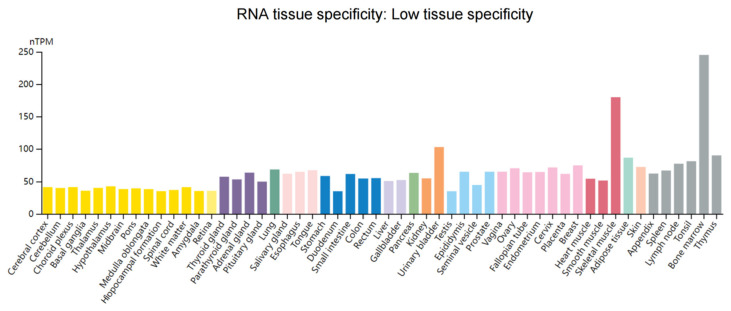
Overview of RNA expression of WTAP in human tissues. Normalized expression (nTPM) of 55 tissue types through the Human Protein Atlas online database (https://www.proteinatlas.org/ (accessed on 25 May 2022)). Data in the database are integrated from RNA-seq dataset of the Human Protein Atlas and RNA-seq dataset of the Genotype-Tissue Expression. Each color corresponds to a tissue with common functional characteristics.

**Figure 5 biomolecules-12-01224-f005:**
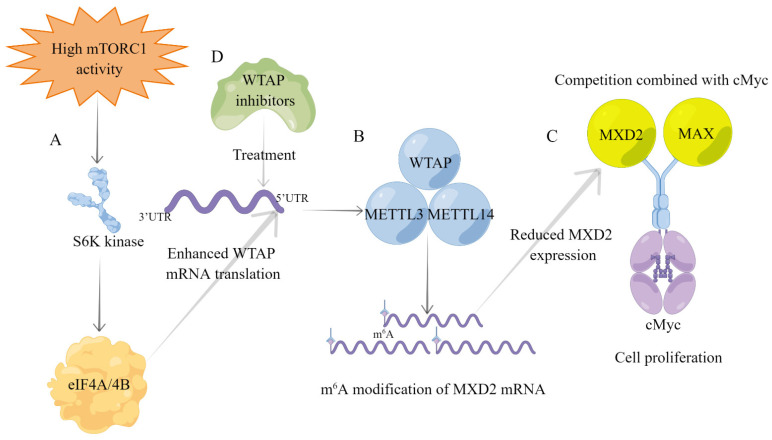
Overview of the mTORC1-WTAP-MXD2-cMyc axis for cancer progression. (**A**) Upregulation of mTORC1 signaling in tumors enhances WTAP mRNA translation via downstream S6K kinase. (**B**) The MMW complex reduces MXD2 expression by m^6^A modification of MXD2 mRNA. (**C**) MXD2 competes with MAX to bind cMyc. The decreased MXD2 expression leads to increased MAX binding to cMyc, thereby promoting tumor cell proliferation. (**D**) WTAP inhibitors inhibit WTAP expression and reduce MXD2 m^6^A modification, thereby inhibiting mTORC1 signaling to activate cancer cell proliferation.

## Data Availability

Not applicable.

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
