# Peer review of "Role of WTAP in Cancer: From Mechanisms to the Therapeutic Potential"

_biomolecules, 2022, doi:10.3390/biom12091224_

Round 1

Reviewer 1 Report

In recent years, a great process has been made in the research of RNA post transcriptional modifications, and hundreds of RNA modifications have been characterized. However, the role of RNA modifications in diverse cellular and biological processes are complicated. Here, Fan Y et al. discussed the current understanding of the WTAP that is required for N6-methyladenosine (m6A). While the authors have made an amount of great work, there are some points that could be further improved.

1.     There are two classes of m6A methylation sites that are present in mammalian mRNAs. They are WTAP-dependent and WTAP-independent m6A sites. The authors should add more discussions of above information in “2. Methyltranferase” section.

2.     The authors discussed the functional roles of WTAP in carcinogenesis. However, are these dysregulated WTAP will affect m6A modification in different diseases? The authors should also discuss their effects on m6A levels.

3.     The authors should add inside horizontal borders to separate the different regulation types in Table1 and different cancer types in Table2. It will help readers to follow the information more conveniently.

Reviewer 2 Report

In the paper “Role of WTAP in cancer: from mechanisms to the therapeutic Potential” Fan et al review the current literature regarding the involvement of WTAP in malignancy. This is a comprehensive review on an interesting topic. However, I frequently found myself consulting the references because I could not understand the idea that the authors were trying to convey. The manuscript definitely needs improvements in style and accuracy. In addition, some parts are redundant while others are not well explained. In general, it feels like the authors did a significant effort to put a lot of information together but failed to expose it in a way that is clear, interesting and friendly to the reader. I have the following suggestions:

The manuscript has many inaccuracies that need to be addressed. Just a couple of examples:

a.       The sentence “METTL3, the first identified methyltransferase” is not accurate. METTL3 was the first m6A methyltransferase described.

b.       The sentence “mTORC1 regulates cMyc signaling through m6A modification of the cMyc repressor MAX dimerization protein” is confusing. It should be clearly explained that mTORC1 enhances translation of WTAP and the subsequent methylation of MXD2. In addition, Figure 5 is wrong as it says transcription instead on translation. The whole paragraph is impossible to understand.

c.       “supervised clustering” should be “unsupervised clustering”

The manuscript needs to improve the style. For example, I frequently noticed a long introduction that takes several sentences until the connection with WTAP is reached. This makes the reading unfocused. Examples:

a.       “MiR-550-1 expression is downregulated in AML and predicts a poor prognosis. Inhibition of AML cell proliferation and promotion of apoptosis was observed by overexpression of miR-550-1. In addition, Hu et al. used a public dataset and in silico analysis to infer that WTAP is a direct target gene downstream of miR-550-1”

b.       “Studies have exhibited that hypoxia generates intratumoral oxygen gradients that contribute to tumor plasticity and heterogeneity and promote tumor cell invasion, metastasis, and drug resistance. WTAP overexpression under normoxic and hypoxic conditions promoted cell proliferation, inhibited apoptosis and induced resistance to cisplatin. Further studies indicated that lncRNA EMS was upregulated in a hypoxic environment and led to the downregulation of miR-758-3p, a downstream target of lncRNA EMS, this further contributed to hypoxia-induced cisplatin resistance in esophageal cancer cells. Mechanistically, lncRNA EMS under hypoxic conditions can act as ceRNA to compete for binding miR-758-3p to upregulate WTAP expression to promote cisplatin resistance in esophageal tumor cells”.

c.       “LncRNA DLGAP1-AS1 is highly expressed in adriamycin-resistant breast cancer cells and promotes the proliferation of drug-resistant cells. Mechanistically, WTAP promotes the stability of lncRNA DLGAP1-AS1 in breast cancer through m6A modifications”.

Many sentences are difficult to understand or lack context. Authors should make an effort to put the reported findings into context. Examples:

a.       The sentence “In addition, the bioinformatic analysis showed that high expression of WTAP was associated with RNA methylation, while low expression of WTAP was associated with high T cell-related immune responses” is out of context. Bioinformatic analysis of what? What does this data really mean? Why is it important?

b.       “Significantly shortened the cycle of apoptosis”. What is the cycle of apoptosis?

c.       “However, in contrast to Dong et al., Wang et al. demonstrated that high WTAP expression predicted a poor prognosis from the perspective of tumor differentiation”. What does the perspective of tumor differentiation mean?

d.       “WTAP overexpression may transduce circ0008399 knockdown-induced promotion of apoptosis”

e.       “WTAP can mediate tumor resistance formation both dependant and independent of its role in m6A modification of methyltransferases”

f.        “In recent years, besides the involvement of m6A in tumor regulation has received wide attention, and its induction of tumor drug resistance also provides new directions for tumor therapy”

g.       “WTAP overexpression enhanced epidermal growth factor receptor (EGFR)-induced EGFR phosphorylation”

h.       “WTAP stabilizes dual specificity phosphatase 6 (DUSP6) mRNA in an m6A-dependent manner and targets it for mRNA methylation”

While some parts of the manuscript are redundant, some interesting topics are not discussed. This is especially relevant in the therapeutic section. For example, it mentions that the WTAP-MXD2-cMyc axis is a potential therapeutic target for mTORC1-driven cancers but it does not discuss on how this could be achieved. Also, METLL3 inhibitors have been recently developed. Could high expression of WTAP be used to predict sensitivity of tumors to these inhibitors? Could other strategies such as disruption of the complex or PROTAC be more promising?

The conclusions and perspectives are a simple summary of the manuscript. The potential role of WTAP in regulating antitumor immunity, tumor microenvironment, or p53 related functions have not been introduced and appear out of the nowhere.

There are many typos that need to be corrected. For example, “WTAP overexpression predicts a poor prognosis” is used several times in the text and should be corrected to WTAP overexpression predicts poor prognosis.

Overall, this review tries to cover many aspects of WTAP function relevant for cancer but fails to report them in a way that is clear to the reader.

Reviewer 3 Report

In the present manuscript, Yongfei Fan et al want to summarize the present knowledge on the

m6A effector WTAP, highlighting its currently known biological functions as well as its

implication in several cancer types.

The authors discuss the role of WTAP in the m6A complex and offer insights on its function,

structure, and expression. Successively, they focus on the involvement of WTAP in cancers and

the possibility to target WTAP. The manuscript is quite readable, detailed and accompanied by

five supporting figures and two table.

To enhance the already great work of the authors, some small changes could be made:

Minor

- The role of WTAP could be discussed more in the introduction

- Fig.1: The quality of the figure could be improved, specifically the icon used to depict

the m6A modification is not clear. Further, there is a typing mistake in “cytoplasm”

- Both Table 1 and Table 2 lack the legend

Round 2

Reviewer 2 Report

The authors have not addressed the main concerns.
